# Autophagy compensates for Lkb1 loss to maintain adult mice homeostasis and survival

Khoosheh Khayati[1], Vrushank Bhatt[1], Zhixian Sherrie Hu[1], Sajid Fahumy[1], Xuefei Luo[1], Jessie Yanxiang Guo[1,2,3]*

[1]Rutgers Cancer Institute of New Jersey, New Brunswick, United States; [2]Department of Medicine, Rutgers Robert Wood Johnson Medical School, New Brunswick, United States; [3]Department of Chemical Biology, Rutgers Ernest Mario School of Pharmacy, Piscataway, United States

**Abstract** Liver kinase B1 (LKB1), also known as serine/threonine kinase 11 (STK11) is the major energy sensor for cells to respond to metabolic stress. Autophagy degrades and recycles proteins, macromolecules, and organelles for cells to survive starvation. To assess the role and cross-talk between autophagy and Lkb1 in normal tissue homeostasis, we generated genetically engineered mouse models where we can conditionally delete *Stk11* and autophagy essential gene, *Atg7*, *respectively or simultaneously,* throughout the adult mice. We found that Lkb1 was essential for the survival of adult mice, and autophagy activation could temporarily compensate for the acute loss of Lkb1 and extend mouse life span. We further found that acute deletion of Lkb1 in adult mice led to impaired intestinal barrier function, hypoglycemia, and abnormal serum metabolism, which was partly rescued by the Lkb1 loss-induced autophagy upregulation via inhibiting p53 induction. Taken together, we demonstrated that autophagy and Lkb1 work synergistically to maintain adult mouse homeostasis and survival.

**\*For correspondence:**
yanxiang@cinj.rutgers.edu

**Competing interests:** The authors declare that no competing interests exist.

## Introduction

Liver kinase B1 (LKB1) is a tumor suppressor, metabolic sensor, and master modulator of AMP-activated protein kinase (AMPK) and mammalian target of rapamycin complex1 (mTORC1) activity, leading to the control of energy metabolism, cell polarity, cell survival, and proliferation (*Corradetti et al., 2004*; *Alessi et al., 2006*; *Nakada et al., 2010*). Heterozygous germline mutations in *Stk11* lead to the development of Peutz-Jeghers syndrome (PJS), an autosomal dominant disease with hamartomatous polyp formation in the gastrointestinal tract (*Giardiello et al., 2000*). Constitutive deficiency of Lkb1 leads to embryonic lethality due to impaired neural tube closure and somitogenesis, mesenchymal tissue cell death, and defective vasculature (*Hardie, 2011*). Specific deletion of Lkb1 in vascular endothelial cells results in dilated embryonic vessels and death at E12.5, which is attributed to the reduced Tgfβ signaling in yolk sac (*Corradetti et al., 2004*; *Londesborough et al., 2008*). Liver-specific deficiency of Lkb1 causes impaired glucose metabolism (*Zhou et al., 2001*). Muscle-specific deletion of Lkb1 results in lower fasting blood glucose and insulin levels, along with increased glucose uptake through muscles (*Koh et al., 2006*). Lkb1 loss in hematopoietic stem cells causes dysfunctional mitochondria, leading to pancytopenia due to reduced levels of ATP, fatty acids, and nucleotides (*Nakada et al., 2010*; *Gan et al., 2010*; *Gurumurthy et al., 2010*). Loss of Lkb1 in intestinal epithelial cells alters immune barrier, changes intestinal colitogenic microbiota, and confers susceptibility to inflammation through reduction of IL-18 via an AMPK-independent pathway (*Liu et al., 2018*). Taken together, tissue-specific knockout studies underscore the importance of Lkb1 in tissue homeostasis, metabolism, and stem cell

maintenance. Somatic *Stk11* mutations are related with a number of human cancers; however, tissue-specific removal of *Stk11* in mice does not necessarily lead to tumor formation (*Ollila and Mäkelä, 2011*).

Autophagy, a highly conserved self-degradative process, plays an essential role in cellular stress responses and survival (*Guo et al., 2013*; *Guo et al., 2016*; *Komatsu et al., 2005*). Yeast cells rely on autophagy to survive nitrogen starvation *Tsukada and Ohsumi, 1993*; neonatal mice depend on autophagy to survive neonatal starvation-induced amino acid depletion (*Komatsu et al., 2005*; *Kuma et al., 2004*) and adult mice requires autophagy to survive starvation (*Komatsu et al., 2005*; *Karsli-Uzunbas et al., 2014*).

Given that both Lkb1 signaling and autophagy play indispensable roles in maintaining tissue energy homeostasis, we began to investigate the interaction of Lkb1 signaling and autophagy in supporting homeostasis of adult mice. We engineered mice to conditionally (Tamoxifen (TAM)-inducible) and systemically delete *Stk11* and *Atg7*, either respectively or simultaneously. Same as previous report (*Karsli-Uzunbas et al., 2014*), systemic *Atg7* ablation led to extensive liver and muscle damage, and neurodegeneration starting at 6 weeks post-deletion, and limited mouse survival to 2.5 months. Surprisingly, we found that adult mice with acute ablation of *Stk11* through whole-body ($Stk11^{-/-}$ mice) died within 25 days and showed upregulated autophagy in most tissues. Moreover, systemic co-deletion of *Stk11* and *Atg7* limited mice survival to 15 days. $Stk11^{-/-}$ mice displayed disruption of intestinal structure and impaired intestinal defense barrier, which was deteriorated by co-deletion with *Atg7*. Supplementation of broad-spectrum antibiotics or systemic deletion of *Trp53* partly rescued the death of the mice with concurrent deletions of *Stk11* and *Atg7*, but not the mice with *Stk11* deletion alone. Serum metabolomics profiling analysis showed that acute short-term deletion of *Atg7* or *Stk11*, respectively, significantly decreased the levels of most essential and non-essential amino acids and some metabolites involved in the tricarboxylic acid (TCA) cycle, urea cycle, and glycolysis. This phenotype was further enhanced in mice with concurrent deletions of *Atg7* and *Stk11*. Taken together, this study reveals a novel role of Lkb1 in the management of tissue homeostasis in adult mouse and the mechanism by which autophagy upregulation temporarily compensates for the acute loss of Lkb1.

## Results

### Acute systemic *Stk11* deletion upregulates autophagy in adult mouse

Tissue-specific *Stk11* knockout studies demonstrate that Lkb1 plays an important role in supporting tissue and organ homeostasis (*Ollila and Mäkelä, 2011*). However, how Lkb1 regulates adult mouse homeostasis is unknown. Autophagy is required to maintain tissue homeostasis and mouse survival in starvation (*Komatsu et al., 2005*; *Karsli-Uzunbas et al., 2014*). Whether and how Lkb1 and Atg7 interact to maintain tissue homeostasis for adult mouse survival remains an open question. To address this, we generated genetically engineered mouse models, in which *Atg7* and *Stk11* were surrounded by flox alleles, whereas the expression of a TAM-regulated Cre-recombinase was manipulated through the ubiquitously expressed ubiquitin C (Ubc) promoter (*Ruzankina et al., 2007*). Four mouse strains were generated: $Ubc\text{-}CreERT2^{/+}$, $Ubc\text{-}CreERT2^{/+};Atg7^{flox/flox}$, $Ubc\text{-}CreERT2^{/+};$ $Stk11^{flox/flox}$, and $Ubc\text{-}CreERT2^{/+};Atg7^{flox/flox};Stk11^{flox/flox}$. Following TAM injections in 8–10 week-old mice, Cre is activated, leading to the deletion of *Atg7* or *Stk11*, and producing a near complete and sustained loss of Atg7 protein ($Atg7^{-/-}$ mice), Lkb1 protein ($Stk11^{-/-}$ mice), or dual deletions of Atg7 and Lkb1 proteins ($Atg7^{-/-};Stk11^{-/-}$ mice) in all tissues (*Figure 1a*). The deletions of Lkb1 and Atg7 throughout the mouse tissues were confirmed by western blot (*Figure 1b*). In addition, accumulation of autophagy substrate p62 was observed in all tissues of the mice with Atg7 ablation ($Atg7^{-/-}$ mice and $Atg7^{-/-};Stk11^{-/-}$ mice) (*Figure 1c*), indicating autophagy blockade.

Given that AMPK is required for autophagy activation, Lkb1 deficiency leads to the loss of AMPK activity and suppression of autophagy (*Lage et al., 2008*). Consequently, studying the role of autophagy in the mice with systemic loss of Lkb1 may be considered counterintuitive. However, studies from us and other groups have shown that AMPK is activated in the mouse *Kras*-driven lung tumors with Lkb1 loss (*Bhatt et al., 2019*; *Eichner et al., 2019*). Indeed, we also observed that autophagy is required for *Kras*-mutant *Stk11*-deficient lung tumorigenesis (*Bhatt et al., 2019*). Instead of Lkb1, AMPK can be activated by calmodulin-dependent kinase kinase (CaMKK) and

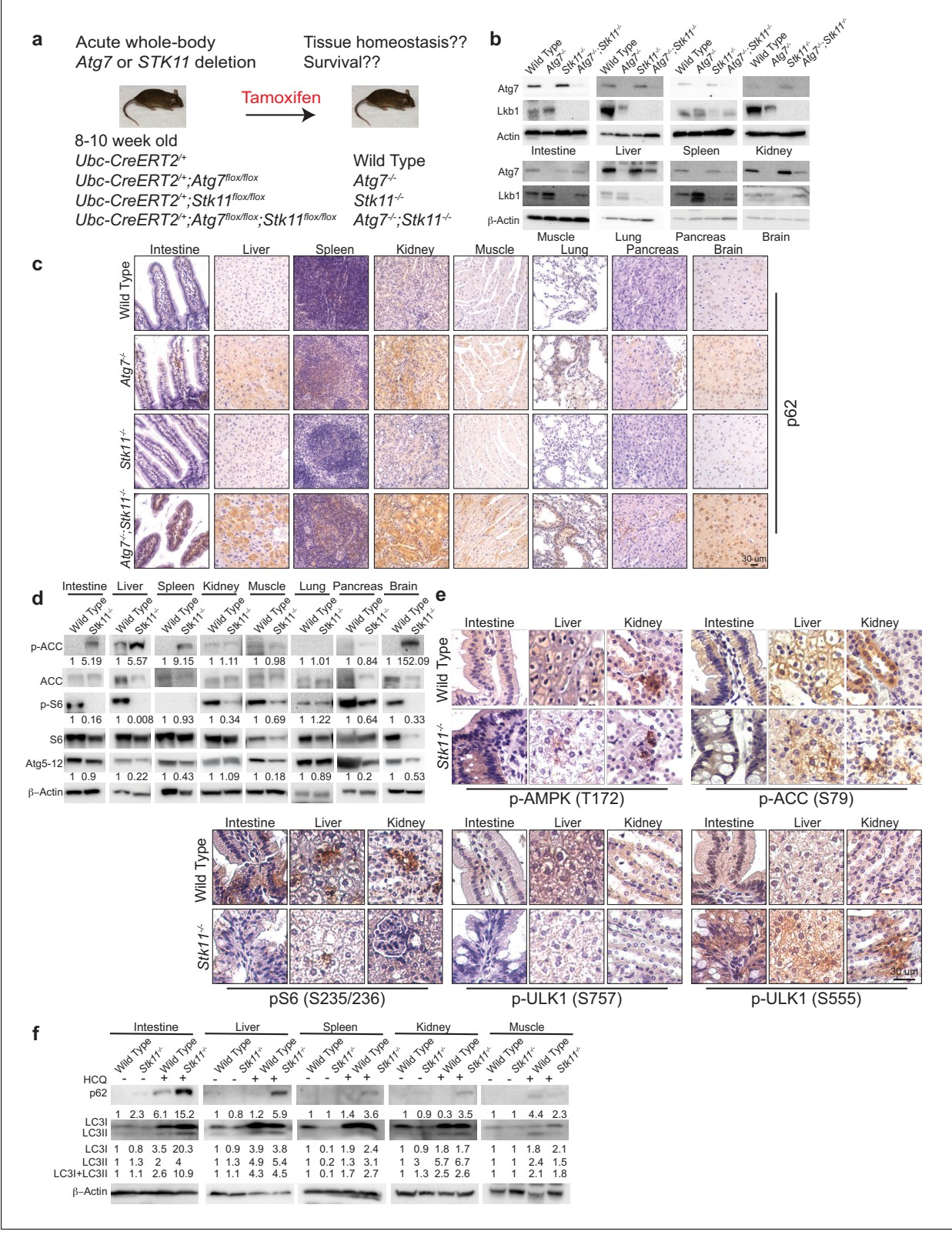

**Figure 1.** Autophagy is upregulated in tissues of *Stk11*-deficient mice. (a) Experimental design for generation of *Atg7⁻/⁻*, *Stk11⁻/⁻*, and *Atg7⁻/⁻;Stk11⁻/⁻* adult mice. (b) Western blotting for Atg7 and Lkb1 of the indicated tissues from WT control, *Atg7⁻/⁻*, *Stk11⁻/⁻*, and *Atg7⁻/⁻;Stk11⁻/⁻* adult mice. β-actin serves as a protein loading control. (c) Representative IHC for p62 of different tissues from WT control, *Atg7⁻/⁻*, *Stk11⁻/⁻*, and *Atg7⁻/⁻;Stk11⁻/⁻* adult mice. (d) Western blotting for pACC (S79), total ACC, pS6 (S235/236), total S6, and Atg5-Atg12 from different tissues of WT control and *Stk11⁻/⁻* mice. β-actin

*Figure 1 continued on next page*

*Figure 1 continued*

serves as a protein loading control. Numbers indicate the quantification of phospho-protein levels normalized to total levels of protein, β-actin and WT control. (e) Representative IHC for pAMPK (Th172), pACC (S79), pS6 (S235/236), and pULK1 (S555 and S757) in different tissues of WT control *and Stk11⁻/⁻* adult mice. (f) Western blotting for p62, LC3I and LC3II in different tissues of WT control and *Stk11⁻/⁻* mice with or without HCQ treatment. β-actin serves as a protein loading control. Numbers indicate the quantification of protein levels normalized to β-actin and WT control.

The online version of this article includes the following figure supplement(s) for figure 1:

**Figure supplement 1.** AMPK is active in Lkb1-deleted tissues.

transforming growth factor beta-activated kinase1 (TAK1) (*Hardie, 2011*; *Sanders et al., 2007*), further leading to autophagy activation. We therefore examined AMPK-mTOR-autophagy axis by assessing the status of phospho (p)-AMPK, phosphorylation of AMPK downstream targets acetyl-CoA carboxylase (ACC) and ULK1, phosphorylation of S6 by mTOC1 substrate S6 Kinase 1, Atg5-Atg12 conjugation and p62 levels (*Alessi et al., 2006*; *Zoncu et al., 2011*). As expected, pAMPK (T172) was observed in the tissues of *Stk11⁻/⁻* mice. Consistent with AMPK activation, phosphorylation of ULK1 (S555) and ACC (S79) was also detected in the tissues of *Stk11⁻/⁻* mice. Most interestingly, although pAMPK was not further increased in the tissues of *Stk11⁻/⁻* mice, pS6 which indicates the activation of mTORC1 signaling was significantly reduced in the tissues of *Stk11⁻/⁻* mice compared with WT mice. With the decrease of pS6, the level of pULK1(S757), a phosphorylation site by mTORC1 (*Egan et al., 2011*; *Kim et al., 2011*), was reduced in the tissues of *Stk11⁻/⁻* mice compared with WT mice (*Figure 1d and e*, and *Figure 1—figure supplement 1a and b*). Moreover, Atg5-Atg12 conjugation was observed in the most tissues of *Stk11⁻/⁻* mice (*Figure 1d*). We further examined the autophagic flux in *Stk11⁻/⁻* mice by administering the mice with hydroxychloroquine (HCQ) that blocks the fusion of autophagosomes with lysosomes (*Yang et al., 2013*). Accumulation of autophagy substrates including both microtubule-associated protein 1A/1B- light chain3 (LC3) I and II along with p62 was observed in most tissues of mice after 5 hr of HCQ treatment. Moreover, HCQ treatment led to a higher accumulation of LC3-II and p62 in *Stk11*-deficient mice compared with WT control mice (*Figure 1f*). Taken together, we demonstrated that autophagy is activated and upregulated in the mice with acute loss of Lkb1.

## Interaction of Lkb1 and autophagy is required for adult mouse survival

It has previously been shown that mice with systemic autophagy ablation have a life span of 2–3 months and the mortality is due to initial Streptococcus infection and eventual neurodegeneration (*Karsli-Uzunbas et al., 2014*), whereas, adult mice with whole-body *Stk11* deletion survive for up to 6 weeks (*Shan et al., 2016*). Given that both Lkb1 signaling and autophagy pathway regulate cellular homeostasis (*Guo et al., 2013*; *Karsli-Uzunbas et al., 2014*; *Lage et al., 2008*; *Gao et al., 2020*), we hypothesize that upregulation of autophagy in *Stk11*-deficient mice might compensate for the acute Lkb1 loss to maintain energy homeostasis for mouse survival. Therefore, we evaluated the overall survival of adult mice with acute deletion of *Atg7* and *Stk11* respectively or in combination. Our data reproduced the survival rate for *Atg7*-deficient mice when *Atg7* was acutely deleted in adult mice (*Figure 2a*; *Karsli-Uzunbas et al., 2014*). The life span of *Stk11*-deficient mice was limited to 3.5 weeks post-TAM administration. Co-deletion of *Atg7* and *Stk11* led to a significant decrease in the survival of *Atg7⁻/⁻;Stk11⁻/⁻* mice compared with *Stk11⁻/⁻* or *Atg7⁻/⁻* mice, resulting 2 weeks of survival (*Figure 2a*).

At 10 days post-TAM injection, compared with WT control mice, there was a loss of body weight in *Stk11⁻/⁻* mice, which was significantly deteriorated by the loss of Atg7 (*Figure 2b*). Hematopoietic-specific *Stk11*-deficient mice died from pancytopenia (*Gan et al., 2010*). We did not observe any differences of red and white blood cell and platelet count among WT control, *Atg7⁻/⁻, and Stk11⁻/⁻* mice; however, the platelet count in *Atg7⁻/⁻;Stk11⁻/⁻* mice was significantly higher than that in WT control mice (*Figure 2—figure supplement 1*).

## Acute autophagy ablation aggravates Lkb1-deficiency-induced loss of secretory cell structure in small intestine

To elucidate the underlying mechanism by which *Atg7* and *Stk11* ablation alone or in concurrent impacts the mouse survival, we examined the histology of different tissues. After short-period (10

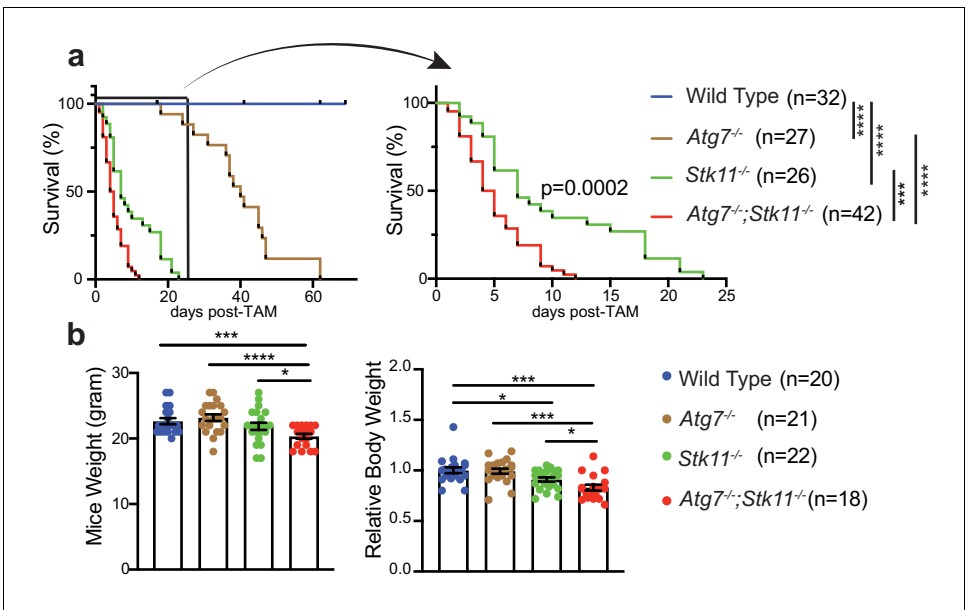

**Figure 2.** Autophagy compensates for acute Lkb1 loss to support the survival of adult mice. (a) Kaplan-Meier survival curve of WT control, *Atg7⁻/⁻*, *Stk11⁻/⁻*, and *Atg7⁻/⁻;Stk11⁻/⁻* adult mice. ***p<0.001, and ****p<0.0001 (log-rank Mantel-Cox test). (b) Left: Body weight was obtained at 10 days post-TAM administration. For the actual weight, 8–10 weeks old mice with original weight range between 20 and 25 g were used. Right: For relative mice weight, each final weight was normalized to its original weight before TAM administration, subsequently normalized to the WT control. Data are mean ± s.e.m. *p<0.05, **p<0.01, and ***p<0.001.

The online version of this article includes the following figure supplement(s) for figure 2:

**Figure supplement 1.** Autophagy and Lkb1 inhibitions do not lead to pancytopenia.

days) of protein deletion, which is before the death of *Atg7⁻/⁻;Stk11⁻/⁻* mice, the damage of tissues was not observed in *Atg7*-deficient mice (***Figure 3—figure supplement 1a***). Moreover, except for the intestine (***Figure 3a***), most of the tissues in both *Stk11⁻/⁻* and *Atg7⁻/⁻;Stk11⁻/⁻* mice were not visibly affected as examined by hematoxylin and eosin (H&E) staining (***Figure 3—figure supplement 1a***). Same as tissue-specific *Stk11* deletion in intestinal-epithelium cells (***Shorning et al., 2009***), enlarged undifferentiated goblet-Paneth cells in the crypt of intestine, including duodenum, jejunum, and ileum, were observed in *Stk11⁻/⁻* mice, which was further exacerbated by the concurrent ablation of *Atg7* in *Atg7⁻/⁻;Stk11⁻/⁻* mice (***Figure 3a***). This observation was further confirmed by Alcian blue staining for goblet cells (***Figure 3b***) and immunohistochemistry (IHC) for lysozyme staining to detect Paneth cells (***Figure 3c***). The decreased intensity of lysozyme staining indicates less frequent and undifferentiated Paneth cells in *Atg7⁻/⁻;Stk11⁻/⁻* mice (***Figure 3c***). In addition to western blot (***Figure 1b***), the deletion of Lkb1 and Atg7 specifically in intestine was also confirmed by IHC (***Figure 3—figure supplement 1b***). In consistent with the induction of autophagy in *Stk11*-deficient mice, LC3-II puncta were observed in the intestine of *Stk11⁻/⁻* mice (***Figure 3—figure supplement 1c***).

To evaluate the role of Lkb1 and autophagy in other components of the small intestinal crypt, we examined the status of stem cells that reside at the bottom of the crypt for regenerating almost all the epithelium cells, including Paneth and goblet cells, enterocytes and tuft cells (***Birchenough et al., 2015***; ***Clevers, 2013***). IHC for olfactomedin4 (OLFM4) (an intestine stem cell marker) shows the lower intensity and frequency of the cells expressing OLFM4 in *Stk11⁻/⁻* and *Atg7⁻/⁻;Stk11⁻/⁻* mice compared with WT control or *Atg7⁻/⁻* mice (***Figure 3d***). However, short-term deletion of *Atg7* alone only impaired Paneth cell formation (***Figure 3c***; ***Wittkopf et al., 2012***).

The structure of the small intestine was extremely damaged by *Lkb1* deletion alone or co-deletions of *Stk11* and *Atg7* (***Figure 3a–d***), which could be due to less regeneration from intestinal stem cells or increased cell death. Cell death in small intestine occurs through apoptosis mostly at the tip of the villi which eventually leads to shedding of dead cells into the lumen (***Blander, 2016***;

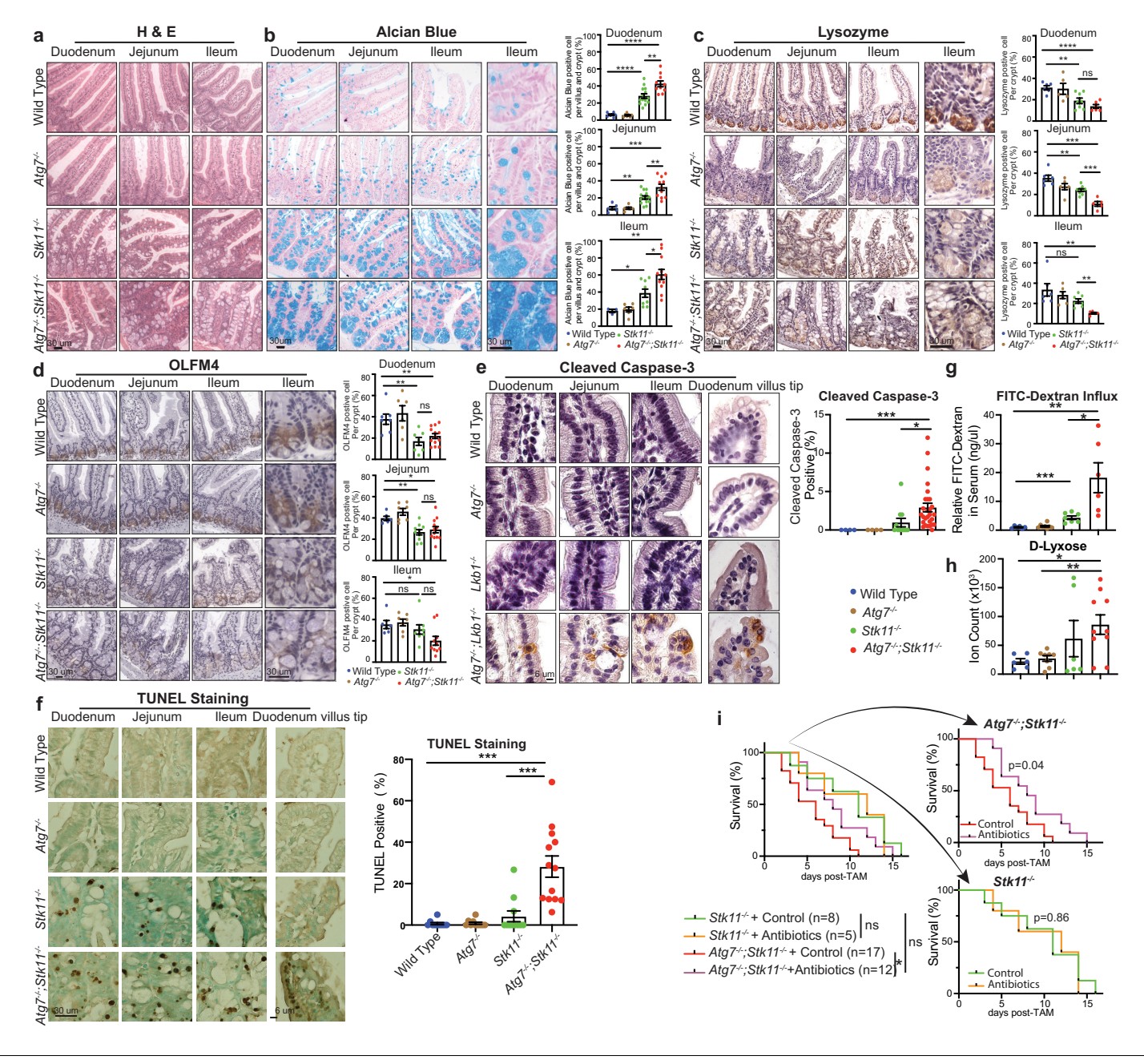

**Figure 3.** Autophagy ablation deteriorates impaired intestinal structure and function caused by acute *Stk11* deletion. (**a**) Representative H&E staining of duodenum, jejunum, and ileum for WT control, *Atg7−/−*, *Stk11−/−*, and *Atg7−/−;Stk11−/−* adult mice. (**b**) Left: Representative alcian blue staining of adult mouse intestine shows the enlargement of mucin-secreting cells in *Stk11−/−* and *Atg7−/−;Stk11−/−* mice. Right: Quantification of alcian blue positive cells in duodenum, jejunum, and ileum. Data are mean ± s.e.m. *p<0.05, **p<0.01, ***p<0.001, and ****p<0.0001. (**c**) Left: Representative IHC for intestinal lysozyme shows decrease of Paneth cell population in *Stk11−/−* and *Atg7−/−;Stk11−/−* crypts. Right: Quantification of lysozyme-positive cells in duodenum, jejunum, and ileum. Data are mean ± s.e.m. **p<0.01, ***p<0.001, ****p<0.0001, and ns means non-significant. (**d**) Left: Representative IHC for OLFM4 of intestine shows the decrease of stem cells with greater extent in *Atg7−/−;Stk11−/−* crypts compared with WT control and *Stk11−/−* mice. Right: Quantification of OLFM4-positive cells in duodenum, jejunum, and ileum. Data are mean ± s.e.m. *p<0.05, **p<0.01, and ns means non-significant. (**e**) Left: Representative IHC for cleaved caspase-3 of intestine delineates increase of cell death in intestine villi and tips of villi in *Atg7−/−;Stk11−/−* compared with WT control and *Stk11−/−* mice. Right: Quantification of cleaved caspase-3. Data are mean ± s.e.m. *p<0.05. (**f**) Left: Representative TUNEL assay in intestine sections shows increase of cell death in intestine villi and tips of villi in *Atg7−/−;Stk11−/−* mice compared with WT control, *Atg7−/−* and *Stk11−/−* mice. Quantification of TUNEL-positive cells in intestine. Data are mean ± s.e.m. ***p<0.001. (**g**) Representative relative FITC-dextran levels in sera of WT control, *Atg7−/−*, *Stk11−/−*, and *Atg7−/−;Stk11−/−* adult mice at 4 hr post-oral gavage of FITC-dextran. Data are mean ± s.e.m. *p<0.05, **p<0.01, ***p<0.001. (**h**) The level of serum D-lyxose in WT control, *Atg7−/−*, *Stk11−/−*, and *Atg7−/−;Stk11−/−* adult mice measured by LC-MS shows an increase of

*Figure 3 continued on next page*

*Figure 3 continued*

D-lyxose in $Atg7^{-/-};Stk11^{-/-}$ sera compared with WT control mice. Data are mean ± s.e.m. *p<0.05, **p<0.01. (i) Kaplan-Meier survival curve of $Stk11^{-/-}$, and $Atg7^{-/-};Stk11^{-/-}$ mice treated with or without broad-spectrum antibiotics (log-rank Mantel-Cox test).

The online version of this article includes the following figure supplement(s) for figure 3:

**Figure supplement 1.** The histology of most mouse tissues and intestinal cell proliferation are not impacted by short-term deletion of Atg7 and Lkb1.

*Günther et al., 2013*). We found that there was a significant increase of apoptotic cell death in the epithelium cells along the villi and at the tip of the villi in $Atg7^{-/-};Stk11^{-/-}$ mice compared with $Stk11^{-/-}$ mice determined by cleaved caspase3 and TUNEL assay (*Figure 3e and f* and *Figure 3—figure supplement 1e*). Intestinal crypt is the region for cell division and migration to upper sites of the villi (*Parker et al., 2017*). Compared with the WT control mice, we did not observe any significant difference in the cell proliferation rate in the crypt of the mice lacking either *Atg7* or *Stk11* alone or in combination (*Figure 3—figure supplement 1d*).

Taken together, we show that Lkb1 is necessary for maintaining the structural integrity of the intestine and that autophagy activation partly compensates for the severe intestinal phenotype induced by the loss of Lkb1.

## Autophagy activation in *Stk11*-deficient mice protects the intestinal epithelium-barrier function

Intestinal epithelium cell-specific deletion of *Stk11* results in an increased susceptibility to dextran sodium sulfate-induced colitis and a definitive shift in the composition of the microbial population in the mouse intestine (*Liu et al., 2018*), suggesting that Lkb1 plays an important role in maintaining the immune barrier function of the intestinal epithelium. Moreover, it has recently been reported that autophagy is essential for the maintenance of Lgr5$^+$ stem cells and regeneration of epithelium barrier during cytotoxic stress (*Trentesaux et al., 2020*). We therefore examined the integrity of intestinal epithelium-barrier in mice with systemic deletion of *Atg7* or *Stk11* alone, or their co-deletion by measuring the Fluorescein Isothiocyanate (FITC)-dextran in-fluxed from the gastrointestinal tract to peripheral circulation. We observed significantly increased levels of serum FITC-dextran in *Stk11*-deficient mice compared with WT control mice, which was further increased by the co-deletion with *Atg7* in $Atg7^{-/-};Stk11^{-/-}$ mice (*Figure 3g*). However, short-term systemic ablation of *Atg7* alone did not impair intestinal epithelium-barrier (*Figure 3g*). This observation suggests that upregulated autophagy by acute *Stk11* deletion is required for the maintenance of intestinal epithelium-barrier.

Loss of intestinal epithelium-barrier causes mice to be susceptible to bacterial infection (*Liu et al., 2018*). We found that the levels of D-lyxose, an aldopentose sugar and a component of the bacterial glycolipids (*Cho et al., 2007*), was significantly increased in $Atg7^{-/-};Stk11^{-/-}$ mice compared with WT control mice (*Figure 3h*), indicating defective bacterial defense. We therefore hypothesized that increased bacterial infection by loss of intestinal epithelium barrier could contribute to the mouse death caused by co-deletions of *Stk11* and *Atg7*. Hence, we treated the $Stk11^{-/-}$ and $Atg7^{-/-};Stk11^{-/-}$ mice with broad-spectrum antibiotics and assessed the mouse survival rate in comparison with untreated ones. Broad-spectrum antibiotics administration significantly extended the survival of $Atg7^{-/-};Stk11^{-/-}$ mice, whereas, did not affect the lifespan of $Stk11^{-/-}$ mice. Moreover, antibiotics treatment led to no significant difference of mouse survival between treated $Atg7^{-/-};Stk11^{-/-}$ mice and un-treated $Stk11^{-/-}$ mice (*Figure 3i*). Thus, one of the potential mechanisms of autophagy activation in response to acute systemic Lkb1 deficiency could be to maintain the survival of mice by preventing bacterial invasion.

## Autophagy activation in *Stk11*-deficient mice prevents p53 activation to maintain mouse survival

Given that autophagy drives an inhibitory role toward p53 activation (*Guo et al., 2013*; *White, 2016*; *Yang et al., 2020*), we expected to observe an increased p53 activation in autophagy-deficient mouse tissues. Indeed, IHC staining for p53 showed that the frequency of nuclear p53 was significantly higher in most tissues of $Atg7^{-/-};Stk11^{-/-}$ mice compared with $Stk11^{-/-}$ mice or WT control mice after short-term deletion of the genes (*Figure 4a* and *Figure 4—figure supplement 1*). This is also accompanied by the significantly increased mRNA levels of p53-targeted downstream genes such as

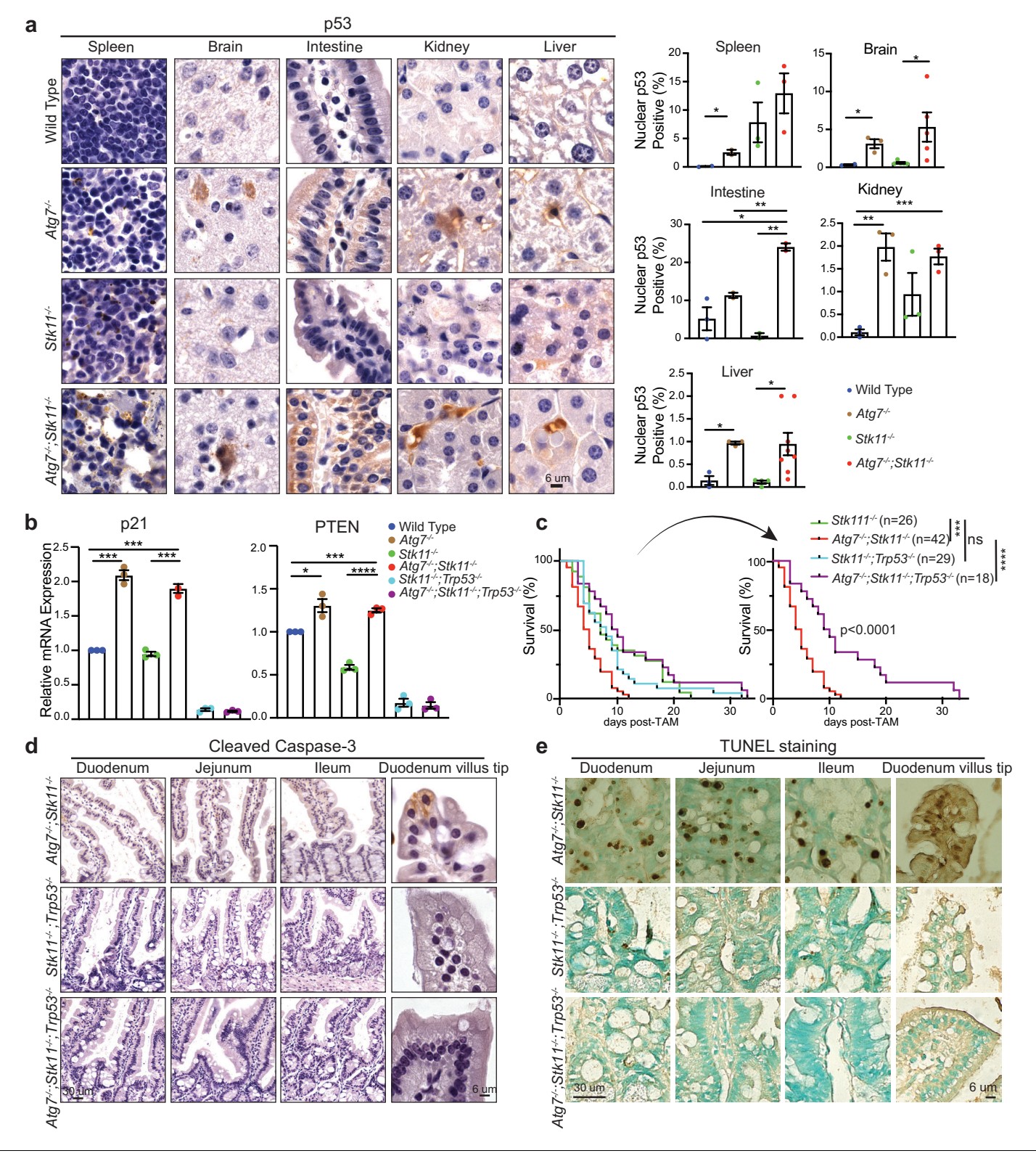

**Figure 4.** p53 deficiency extends the life span of *Atg7⁻/⁻;Stk11⁻/⁻* mice.  (**a**) Left: Representative IHC for p53 in different tissues of WT control, *Atg7⁻/⁻*, S*tk11⁻/⁻*, and *Atg7⁻/⁻;Stk11⁻/⁻* adult mice shows an increase of nuclear p53 in *Atg7*-ablated tissues. Right: Bar graphs represent the quantification of nuclear p53 in different tissues. Data are mean ± s.e.m. *p<0.05, **p<0.01, and ***p<0.001. (**b**) Quantitative real-time PCR of *Cdkn1a* (p21) and *PTEN* for intestine tissues of WT control, *Atg7⁻/⁻*, S*tk11⁻/⁻* and *Atg7⁻/⁻;Stk11⁻/⁻* adult mice. Data are mean ± s.e.m. *p<0.05, ***p<0.001, and ****p<0.0001. (**c**)
*Figure 4 continued on next page*

*Figure 4 continued*

Kaplan-Meier survival curve of *Stk11*[-/-], *Atg7*[-/-];*Stk11*[-/-], *Stk11*[-/-];*Trp53*[-/-], and *Atg7*[-/-];*Stk11*[-/-];*Trp53*[-/-] adult mice. \*\*\*p<0.001, \*\*\*\*p<0.0001 and ns: non-significant (log-rank Mantel-Cox test). (d) Representative IHC for cleaved caspase-3 of intestine from *Atg7*[-/-];*Stk11*[-/-], *Atg7*[+/+];*Stk11*[-/-];*Trp53*[-/-], and *Atg7*[-/-];*Stk11*[-/-];*Trp53*[-/-] adult mice. (e) Representative TUNEL assay of intestine from *Atg7*[-/-];*Stk11*[-/-], *Atg7*[+/+];*Stk11*[-/-];*Trp53*[-/-] and *Atg7*[-/-];*Stk11*[-/-];*Trp53*[-/-] adult mice.

The online version of this article includes the following figure supplement(s) for figure 4:

**Figure supplement 1.** p53 is activated in the absence of autophagy.

---

*p21* and phosphatase and tensin homolog (*PTEN*) in *Atg7*[-/-];*Stk11*[-/-] mice compared with *Stk11*[-/-] mice or WT control mice (*Figure 4b*). Accordingly, we tested the hypothesis that activation of autophagy by acute *Stk11* ablation could prevent mouse death by inhibiting p53 activation. To address this, two new cohorts of mice were generated: *Ubc-CreERT2*[/+];*Stk11*[flox/flox];*Trp53*[flox/flox], and *Ubc-CreERT2*[/+];*Atg7*[flox/flox];*Stk11*[flox/flox];*Trp53*[flox/flox]. TAM administration can cause concurrent deletion of *Stk11* and *Trp53* (*Stk11*[-/-];*Trp53*[-/-] mice) or *Atg7*, *Stk11*, and *Trp53* (*Atg7*[-/-];*Stk11*[-/-];*Trp53*[-/-] mice) throughout the whole body. We evaluated the role of *Trp53* deletion in mouse survival by comparing with *Stk11*[-/-] mice and *Atg7*[-/-];*Stk11*[-/-] mice that have intact *Trp53*. Systemic *Trp53* ablation had no effect on the survival of *Stk11*-deficient mice. However, whole-body ablation of *Trp53* significantly extended the life span of *Atg7*[-/-];*Stk11*[-/-] mice (*Figure 4c*). Moreover, the apoptotic cell death in intestine was abolished when p53 was deleted (*Figure 4d and e*). Thus, when *Stk11* is acutely ablated throughout the adult mice, autophagy inhibits p53 activation to temporarily extend the mouse life span.

## Lkb1 and Atg7 are required to maintain adult mice homeostasis

To further elucidate the underlying mechanism of Lkb1 and Atg7 in supporting adult mice survival, we performed serum metabolomics in fasting state after short-term deletion of the genes. Of the 90 metabolites we examined, we found that acute deletion of *Atg7* or *Stk11* alone significantly decreased the levels of most essential and non-essential amino acids and some metabolites involved in urea cycle and glycolysis (*Figure 5a–d*). Interestingly, we found that the reduced levels of TCA cycle intermediates were only observed in the absence of *Atg7* (*Atg7*[-/-] mice and *Atg7*[-/-];*Stk11*[-/-] mice), but not in *Stk11*-deficient mice (*Figure 5b*). We further found that in both fasted state (*Figure 5d*) and fed state (*Figure 5e*), blood glucose level was significantly lower in *Stk11*[-/-] and *Atg7*[-/-];*Stk11*[-/-] mice compared with WT control mice. Following that, we evaluated the serum insulin levels in all four groups of mice. The insulin levels in *Stk11*[-/-] and *Atg7*[-/-];*Stk11*[-/-] mice were decreased with the same trend as glucose compared with WT control mice (*Figure 5f*). We also performed a metabolomics profiling analysis of the intestinal tissue (ileum) which showed significantly impaired histology and function in both *Stk11*[-/-] and *Atg7*[-/-];*Stk11*[-/-] mice (*Figure 3*). The alteration of metabolic pathways in the intestine metabolomics profiling due to *Atg7* and *Stk11* ablation was consistent with the change in the serum metabolomics profiling; that is, the loss of *Stk11* alone or together with *Atg7* ablation resulted in the decreased levels of certain intermediates involved in the amino acid metabolism, TCA cycle, urea cycle, and glycolysis (*Figure 5—figure supplement 1*). Thus, autophagy synergizes with Lkb1 to maintain host homeostasis in the adult mice.

## Discussion

In this study, we demonstrated the intermingled essential and systemic roles of Lkb1 and autophagy in the maintenance of mouse homeostasis and survival via conditional whole-body deletion of *Stk11* and *Atg7* in adult mice (*Figure 6*). We found that acute Lkb1 loss led to damaged intestinal epithelium barrier and increased infection, and alteration in metabolic pathways necessary for maintaining host homeostasis, which was partially rescued by autophagy activation via inhibiting p53 induction. Thus, autophagy upregulation compensates for the acute Lkb1 loss to temporarily support the survival of adult mice.

An accumulating body of evidence suggests that Lkb1 phosphorylates AMPK and activates autophagy in response to energy crises (*Corradetti et al., 2004*; *Alessi et al., 2006*; *Nakada et al., 2010*; *Hardie, 2011*; *Lage et al., 2008*). However, in addition to Lkb1, CaMKK, and TAK1 also

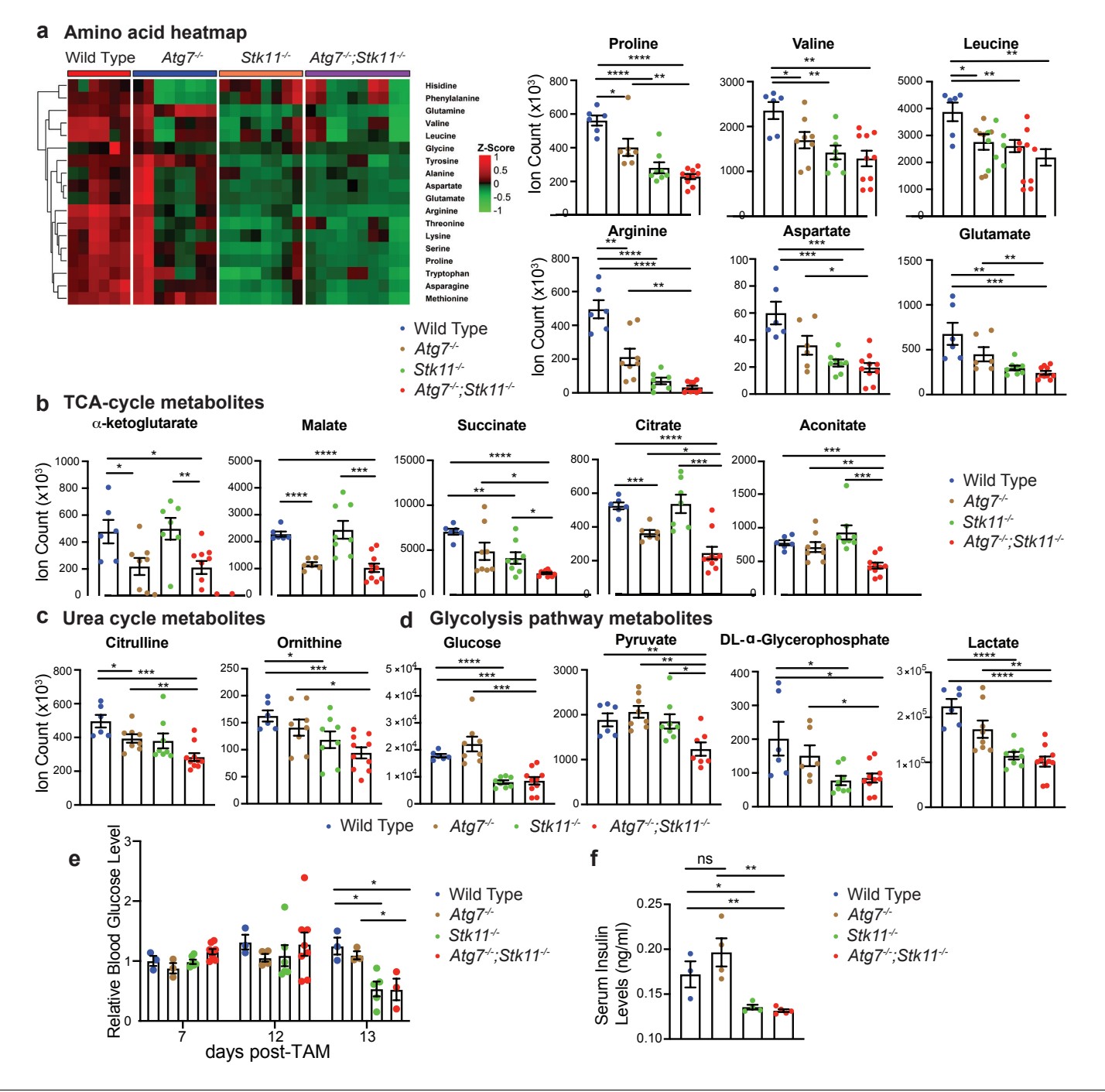

**Figure 5.** Major metabolic pathways are disturbed in *Stk11⁻/⁻* and *Atg7⁻/⁻;Stk11⁻/⁻* mice. (**a**) Left: Representative heat map of all the amino acids in sera of WT control, *Atg7⁻/⁻*, *Stk11⁻/⁻*, and *Atg7⁻/⁻;Stk11⁻/⁻* adult mice compared with WT control mice at fasting state. Right: Bar graphs show the levels of amino acids that are significantly decreased in *Stk11⁻/⁻* and *Atg7⁻/⁻;Stk11⁻/⁻* mice sera compared with WT control mice. Data are mean ± s.e.m. *p<0.05, **p<0.01, ***p<0.001, and ****p<0.0001. (**b-d**). Metabolites that are significantly decreased in the sera of *Stk11⁻/⁻* and *Atg7⁻/⁻;Stk11⁻/⁻* mice compared with WT control mice at fasting state. Data are mean ± s.e.m. *p<0.05, **p<0.01, ***p<0.001, and ****p<0.0001. (**e**) Relative blood glucose levels of WT control, *Atg7⁻/⁻*, *Stk11⁻/⁻* and *Atg7⁻/⁻;Stk11⁻/⁻* adult mice normalized to WT control mice at fed state for the indicated time course after first TAM injection. Data are mean ± s.e.m. *p<0.05. (**f**) Quantification of serum insulin levels of WT control, *Atg7⁻/⁻*, *Stk11⁻/⁻* and *Atg7⁻/⁻;Stk11⁻/⁻* adult mice at fasted state at 10 days post-deletion. Data are mean ± s.e.m. *p<0.05, **p<0.01, ns: non-significant.

The online version of this article includes the following figure supplement(s) for figure 5:

**Figure supplement 1.** Acute loss of Lkb1 alone or together with Atg7 altered the levels of metabolites in intestine.

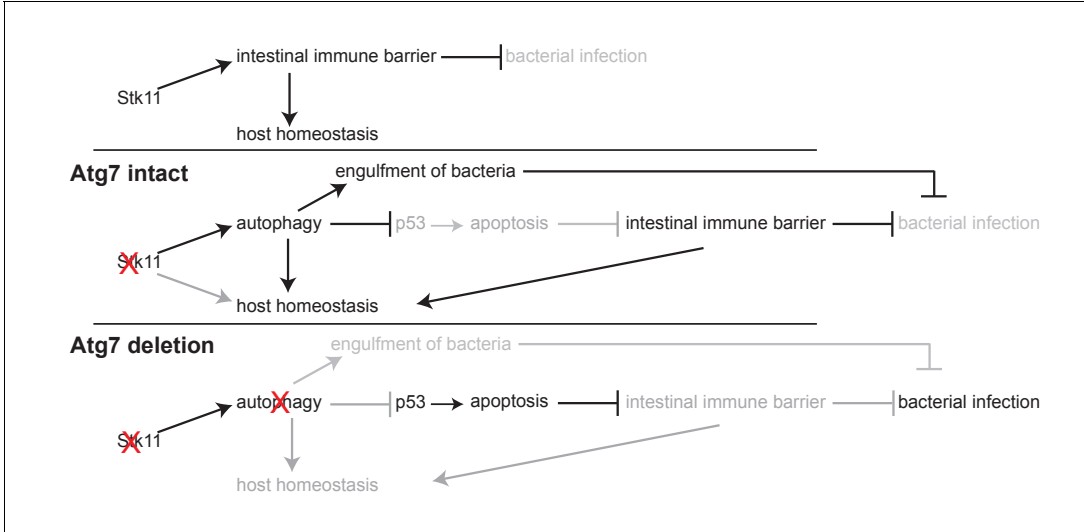

**Figure 6.** Mechanism by which Lkb1 interacts with autophagy to support adult mice homeostasis and survival. Loss of Lkb1 causes hypoglycemia, impaired intestinal epithelium barrier integrity, increased general infection, and disturbed host sera metabolism. When autophagy is intact, these dysfunctions are temporarily compensated by autophagy upregulation, partly through preventing p53 activation. However, autophagy deficiency further exacerbates the dysfunctions induced by *Stk11* deletion, thereby accelerating mouse death.

trigger AMPK activation (*Sanders et al., 2007*). Here, we confirmed AMPK activation and upregulation of autophagy in *Stk11*-deficient mice, suggesting that Lkb1-AMPK-mTORC1-autophagy axis may not be the only or the critical effector of Lkb1-mediated maintenance of adult mice homeostasis in our setting. Ribosomal protein S6 phosphorylation is commonly used as a readout of mTORC1 signaling (*Ruvinsky et al., 2005*). The level of pS6 was significantly reduced in the tissues of *Stk11*[-/-] mice compared with WT control mice. mTORC1 activity can be stimulated by growth factors and cellular energy through inhibition of the tuberous sclerosis complex (TSC) complex (TSC1-TSC2-TBC1D7), a negative regulator of mTORC1. However, amino acids can also signal to mTORC1 independently of the TSC complex via sensors localized on the lysosomal membrane (*Zoncu et al., 2011*; *Jewell et al., 2015*; *Bar-Peled and Sabatini, 2014*; *Sancak et al., 2010*; *Shimobayashi and Hall, 2016*). We showed that systemic ablation of Lkb1 and autophagy led to the depletion of amino acids and other essential metabolites in serum, including leucine, arginine, and methionine, which are essential for mTORC1 activation (*Zoncu et al., 2011*; *Jewell et al., 2015*; *Bar-Peled and Sabatini, 2014*; *Sancak et al., 2010*; *Shimobayashi and Hall, 2016*; *Wang et al., 2015*; *Saxton et al., 2016*; *Wolfson et al., 2016*; *Gu et al., 2017*; *Abu-Remaileh et al., 2017*; *Chantranupong et al., 2016*; *Wolfson et al., 2017*; *Wyant et al., 2017*). Therefore, systemic metabolic collapse could lead to mTORC1 inactivation. This is consistent with a previous study that Lkb1-mediated energy metabolism is largely independent of Lkb1 regulation of AMPK signaling in mouse hematopoietic stem cells (*Gan et al., 2010*). Our study suggests that inactivation of mTORC1 in *Stk11*[-/-] mice could be obtained through both AMPK-dependent and independent pathways, leading to autophagy activation to compensate for acute Lkb1 loss.

Homozygous deletion of *Stk11* in mouse is embryo-lethal (*Hardie, 2011*), demonstrating an important role of Lkb1 in embryogenesis. Here, we reported an indispensable role of Lkb1 in maintaining the homeostasis and survival of adult mice. Mice carrying one inactivated allele of *Lkb1* (*Stk11*[+/−]) recapitulate PJS and die at 11 months after birth due to the development of intestinal polyps (*Udd et al., 2010*). Hematopoietic stem cell-specific deletion of *Stk11* led to limited mouse survival for up to 28 days due to pancytopenia (*Gan et al., 2010*). However, in our study, although mice died within 3 weeks after *Stk11* ablation, no obvious reduction in different types of blood cells was observed. Instead, serum metabolomics analysis showed significant reduction in most of the essential and non-essential amino acids, certain metabolites related to the urea cycle and glycolysis in *Stk11*[-/-] mice compared with WT mice. Thus, the impaired systemic homeostasis by acute Lkb1 loss may be responsible for the death of *Stk11*[-/-] mice.

Lkb1 is involved in the development and maintenance of the goblet and Paneth cells (*Shorning et al., 2009*). Specific deletion of *Stk11* in intestinal stem cells leads to increased expression of pyruvate dehydrogenase kinase4 and reduced oxygen consumption, which reduces the population of stem cells and increase the levels of secretory cell number (*Gao et al., 2020*). The impaired goblet and Paneth cells are also characteristics of PJS in humans (*Udd et al., 2010*). The whole-body deletion of *Stk11* in our mice recapitulates the intestinal phenotype observed previously. Additionally, we found that the impaired intestinal structure was further exaggerated when *Atg7* is concurrently deleted with *Stk11*. The major function of Goblet cells is attributed to the secretion of mucus to provide the epithelium immune barrier against bacterial invasion from the intestinal lumen (*Birchenough et al., 2015*). Moreover, Lkb1 loss leads to the deficiency of immune barrier through an AMPK-independent pathway which is required for the production of antimicrobial IL-18 (*Liu et al., 2018*). Here we demonstrated that the integrity of intestinal-epithelium barrier and function are damaged with a greater extent in $Atg7^{-/-};Stk11^{-/-}$ mice compared with $Stk11^{-/-}$ mice, leaving the mice vulnerable to infection. The role of autophagy in microbial defense is well established (*Gutierrez et al., 2007*; *Tan et al., 2009*; *Terebiznik et al., 2009*). One of the main roles of intestinal epithelium autophagy is to engulf invaded bacteria from extra intestinal sites, which is implied by innate immune adaptor protein MyD88 (*Benjamin et al., 2013*). Autophagy is also crucial for the function of Paneth cells (*Deretic and Levine, 2009*; *Cadwell et al., 2008*), and autophagy deficiency in intestinal epithelium results in inflammatory bowel disease and abnormal Paneth cell formation (*Wittkopf et al., 2012*). We found that broad-spectrum antibiotics supplementation could partially rescue the death of $Atg7^{-/-};Stk11^{-/-}$ mice, but not $Stk11^{-/-}$ mice, suggesting the role of autophagy in preventing infection-related death. Here, we demonstrated for the first time the role of autophagy in preventing infection-related death induced by Lkb1 loss. However, the cause of $Stk11^{-/-}$ mice death could be systemic catastrophe, rather than bacterial infection alone based on serum metabolomics profiling with the depletion of amino acids, essential metabolites and low glucose. p53 is activated upon autophagy deletion, leading to increased apoptotic rates in mouse liver and brain, impairing the overall mouse homeostasis (*Yang et al., 2020*). A recent study shows that *Atg7* specific deletion in Lgr5+ epithelium cells promotes p53-induced apoptosis, leading to impaired integrity of intestinal barrier during stress (*Trentesaux et al., 2020*). Our data was in line with these recent findings, where we found that whole body deletion of *Trp53* significantly rescued the survival of $Atg7^{-/-};Stk11^{-/-}$ mice. Conversely, *Trp53* deletion had no effect on the life span of $Stk11^{-/-}$ mice, showing that autophagy deletion in $Atg7^{-/-};Stk11^{-/-}$ mice affects mouse survival through activation of p53. Loss of intestinal barrier integrity is ascribed to the imbalance between cell proliferation in the crypt and cell migration toward the tip of villi where eventually cells undergo apoptosis (*Parker et al., 2017*). We observed increased apoptotic cell death in the villus epithelial cells of intestine in $Atg7^{-/-};Stk11^{-/-}$ mice compared with $Stk11^{-/-}$ mice, which was rescued by systemic p53 ablation. However, the rate of cell proliferation in the crypt between $Atg7^{-/-};Stk11^{-/-}$ and $Stk11^{-/-}$ mice showed no significant difference. Thus, autophagy upregulation induced by acute Lkb1 loss plays an important role in maintaining a balance between cell proliferation and cell death in the intestine, presumably by inhibiting p53 induction.

Reduction of serum glucose and insulin levels was observed in both $Stk11^{-/-}$, and $Atg7^{-/-};Stk11^{-/-}$ mice, which is in parallel with muscle-specific *Stk11*-deficient mice, which showed decreased blood glucose and insulin levels due to increased uptake of glucose through muscles (*Koh et al., 2006*). However, it will also be interesting to clarify whether systemic ablation of Lkb1 can damage liver gluconeogenesis and cause hypoglycemia. In addition, the decrease in insulin may be due to the response to hypoglycemia, which does not rule out the possibility of abnormal pancreatic function, although the damage of pancreas was not visible by histology examination (*Figure 3—figure supplement 1a*). Most of the intermediates associated with amino acid metabolism, urea cycle, TCA cycle, and glycolysis were significantly decreased in serum by acute *Stk11* ablation. Certain metabolites even showed higher extent of reduction in $Atg7^{-/-};Stk11^{-/-}$ mice compared with $Stk11^{-/-}$ mice. Moreover, the changes of metabolomics profiling in the intestine is consistent with that in serum, suggesting that this alteration occurs throughout the whole body, not a specific tissue. Taken together, both autophagy and Lkb1 are essential to maintain the host metabolism in adult mice.

Autophagy upregulation extended the lifespan of $Stk11^{-/-}$ mice. However, it failed to restore their homeostasis and survival. Moreover, p53-deficiency only improved the symptoms imposed by Atg7-deficiency, but not ameliorate any defects caused by Lkb1-deficiency alone. Lkb1 is known as a hub

for maintenance of cellular polarity, structure, proliferation, and metabolism (*Ollila and Mäkelä, 2011*), which could be autophagy-independent. For example, Lkb1 deficiency leads to alteration of energy metabolism in hematopoietic stem cells independent of mTOR/autophagy pathway (*Gan et al., 2010*). Unlike tissue-specific *Stk11* knockout alone, systemic Lkb1 ablation in adult mice may cause severe host metabolic disorders due to multiple organ failure, which may not be observed by histology H&E staining alone. Therefore, in the future, more functional assays are required to fully understand the autophagy-independent pathways impaired by Lkb1 loss and how it contributes to the dysfunctions caused by Lkb1 deficiency.

# Materials and methods

## Key resources table

| Reagent type (species) or resource | Designation | Source or reference | Identifiers | Additional information |
|---|---|---|---|---|
| Antibody | Atg7 (Rabbit polyclonal) | Sigma Aldrich | Cat# A2856 | IHC: (1:400) WB: (1:2000) |
| Antibody | Lkb1 (mouse monoclonal) | Sant Cruz Biotechnology | Cat# sc-32245 | IHC: (1:50) WB: (1:500) |
| Antibody | P62 (Guinea pig)/ (Rabbit polyclonal) | Enzo Life Sciences/ American Research Products | Cat# PW9860-0100/ 03-GP62-C | IHC: (1:1000) WB: (1:2500) |
| Antibody | p-AMPK$^{Th172}$ (Rabbit polyclonal) | Cell Signaling Technology | Cat#: 2535S | IHC: (1:100) |
| Antibody | p-ACC$^{S79}$ (Rabbit polyclonal) | Cell Signaling Technology | Cat#: 3661 | IHC: (1:800) WB: (1:1000) |
| Antibody | ACC (Rabbit polyclonal) | Cell Signaling Technology | Cat#: 3676 | WB: (1:1000) |
| Antibody | p-S6$^{S235/236}$ (Rabbit polyclonal) | Cell Signaling Technology | Cat#: 4858 | IHC: (1:300) WB: (1:2000) |
| Antibody | S6 (Rabbit polyclonal) | Cell Signaling Technology | Cat#: 2217 | WB: (1:1000) |
| Antibody | p-ULK1$^{S555}$ (Rabbit polyclonal) | Cell Signaling Technology | Cat#: 5869 | IHC: (1:100) |
| Antibody | p-ULK1$^{S757}$ (Rabbit polyclonal) | Cell Signaling Technology | Cat. #: 14202 | IHC: (1:800) |
| Antibody | LC3 (Rabbit polyclonal) | Nano Tolls | Cat. #: LC3-5F10 | IHC: (1:100) WB: (1:4000) |
| Antibody | Ki67 (Rabbit polyclonal) | Abcam | Cat. #: ab-15580 | IHC: (1:400) |
| Antibody | Cleaved Caspase3 (Rabbit polyclonal) | Cell Signaling Technology | Cat. #: 9661S | IHC: (1:250) |
| Antibody | OLFM4 (Rabbit polyclonal) | Cell Signaling Technology | Cat. #: 39141 | IHC: (1:2000) |
| Antibody | Lysozyme (Rabbit polyclonal) | Aligent | Cat. #: A0099 | IHC: (1:2000) |
| Antibody | P53 (mouse monoclonal) | Novus Biologicals | Cat. #: NB200-103SS | IHC: (1:300) |

*Continued on next page*

*Continued*

| Reagent type (species) or resource | Designation | Source or reference | Identifiers | Additional information |
|---|---|---|---|---|
| Antibody | Atg5 (Rabbit polyclonal) | Abcam | Cat. #: ab108327 | WB: (1:500) |
| Antibody | β-actin (mouse monoclonal) | Sigma Aldrich | Cat. #: A1978 | WB: (1:2000) |
| Commercial assay or kit | TUNEL assay | Abcam | ab206386 | |
| Software, algorithm | Prism GraphPad | Prism 8 | RRID:SCR_002798 | |
| Software, algorithm | Adobe Illustrator | CC2020 | RRID:SCR_010279 | |
| Mouse genotype | Ubc-CreERT2 | Jackson Laboratory | | |
| Mouse genotype | *Stk11$^{flox/flox}$* | Jackson Laboratory | | |
| Mouse genotype | *Trp53$^{flox/flox}$* | Jackson Laboratory | | |
| Mouse genotype | *Atg7$^{flox/flox}$* | *Komatsu et al., 2005* | | |

## Mice

All animal experiments were performed in compliance with Rutgers Animal Care and Use Committee (IACUC) guidelines. *Ubc-CreERT2* mice (*Ruzankina et al., 2007*) (Jackson Laboratory) were cross-bred with *Atg7$^{flox/flox}$* mice (*Komatsu et al., 2005*), *Stk11$^{flox/flox}$* mice (Jackson Laboratory) and *Trp53$^{flox/flox}$* mice (Jackson Laboratory) to generate *UbcCreERT2$^{/+}$;Atg7$^{flox/flox}$* mice, *UbcCreERT2$^{/+}$;Stk11$^{flox/flox}$* mice, *UbcCreERT2$^{/+}$;Atg7$^{flox/flox}$;Stk11$^{flox/flox}$* mice, *UbcCreERT2$^{/+}$;Stk11$^{flox/flox}$;Trp53$^{flox/flox}$* mice and *UbcCreERT2$^{/+}$;Atg7$^{flox/flox}$;Stk11$^{flox/flox}$;Trp53$^{flox/flox}$* mice.

For the acute deletion of *Atg7*, *Stk11*, and *Trp53*, TAM (200 µl of suspended solution per 20 g body weight) was delivered to 8- to 10-week-old adult mice through intraperitoneal (IP) injections every 3 days for four times. In the analysis of the survival curve, day 1 is the day of the third shot when deletion of genes was obtained.

To examine autophagy flux in adult mice, HCQ (100 mg/kg) was applied to the mice through IP injection.

To examine the effect of broad-spectrum antibiotics on the survival of *Stk11$^{-/-}$* or *Atg7$^{-/-}$;Stk11$^{-/-}$* mice, broad-spectrum antibiotics Baytril (2.27% enrofloxacin) (5 mg/kg) was injected to the mice via IP twice per day.

Body weight was obtained at 10 days post-TAM administration. For relative mice weight, each final weight was normalized to its original weight before TAM administration, subsequently normalized to the WT control.

## Serum assays

Blood glucose was measured using a True2Go glucose meter (Nipro Diagnostics), and liquid chromatography–mass spectrometry (LC-MS) metabolomics analysis (mentioned below). Serum insulin levels were assessed with an ultra-sensitive mouse insulin (Crystal Chem Inc, 90080) kit.

## Metabolomics analysis by LC-MS

Tissue or serum metabolites extracted using methanol:acetonitrile:water (40:40:20) (with 0.5% formic acid solution for tissue metabolite extraction and without formic acid for serum metabolite extraction) followed by neutralization with 15% ammonium bicarbonate were used for LC-MS, as described previously (*Guo et al., 2016*). Samples were subjected to reversed-phase ion-pairing chromatography coupled by negative mode electrospray ionization to a stand-alone orbitrap mass spectrometer (Thermo Fisher Scientific).

## Intestinal permeability analysis

In vivo intestinal permeability was measured by FITC-dextran (Sigma Aldrich) gavage experiment. Mice were deprived from water overnight before oral gavaging with FITC-dextran at 44 mg/100 g body weight. Subsequently, water was provided after gavage and blood samples were collected by cardiac puncture at 4 hr post-gavage. Sera were collected after centrifuging blood at 10,000 rpm for 10 min using 1.5 mL heparin-lithium coat tubes. Fluorescence intensity of the serum was measured, and the concentration of FITC-dextran was assessed according to the standard curve generated by the serial dilution of FITC-dextran (*Liu et al., 2018*).

## Histology, immunohistochemistry, and TUNEL assay

Paraffin-embedded tissue sections were prepared as described previously (*Guo et al., 2013*) for H and E, and IHC staining. Antibodies utilized for IHC were Atg7 (Sigma Aldrich, A2856, RRID:AB_1078239), Lkb1 (Santa Cruz Biotechnology, sc-32245, RRID:AB_627890), p62 (Enzo Life Sciences, PW9860-0100, RRID:AB_2877676), p-AMPK$^{Th172}$ (Cell Signaling, 2535S, RRID:AB_331250), p-ACC$^{S79}$ (Cell Signaling, 3661, RRID:AB_330337), p-S6$^{S235/236}$ (Cell Signaling, 4858, RRID:AB_916156), p-ULK1$^{S555}$ (Cell Signaling, 5869, RRID:AB_10707365), p-ULK1$^{S757}$ (Cell Signaling, 14202, RRID:AB_2665508), LC3 (Nano Tools, LC3-5F10, RRID:AB_2722733), Ki67 (Abcam, ab-15580, RRID:AB_443209), cleaved caspase-3 (Cell Signaling, 9661S), OLFM4 (Cell Signaling, 39141, RRID:AB_2650511), lysozyme (Agilent, A0099, RRID:AB_2341230), and p53 (Novus Biologicals, NB200-103SS, RRID:AB_2877680). Paraffin embedded tissue sections were used for the TUNEL assay by means of the HRP-DAB TUNEL staining kit (ab206386) and the slides were counterstained by methyl blue following the protocol provided by the TUNEL staining kit. For the quantification of IHC and TUNEL assay, tissues were analyzed by quantifying at least 10 images at 20x magnification. A minimum of 200 cells were scored for each image.

## Western blotting

Tissues were snap-frozen in liquid nitrogen, ground using Cryomill in liquid nitrogen at 25 Hz for 2 min, and then lysed in Tris lysis buffer (1M Tris-HCl, 1M NaCl, 0.1M EDTA, 10% NP40). Protein concentrations were measured using the Bio-Rad BCA reagent. Samples were probed with antibodies against Atg7 (Sigma Aldrich, A2856, RRID:AB_1078239), Lkb1 (Santa Cruz Biotechnology, sc-32245, RRID:AB_627890), LC3 (Novus Biologicals, NB600-1384, RRID:AB_669581), p62 (American Research Products, 03-GP62-C, RRID:AB_1542690), p-ACC$^{S79}$ (Cell Signaling, 3661, RRID:AB_330337), ACC (Cell Signaling, 3676, RRID:AB_2219397), Atg5 (Abcam, ab108327, RRID:AB_2650499), p-S6$^{S235/236}$ (Cell Signaling, 4858, RRID:AB_916156), S6 (Cell Signaling, 2217, RRID:AB_331355), and β-actin (Sigma Aldrich, A1978, RRID:AB_476692). Western blots were quantified with Image J (National Institutes of Health). The intensities of bands were used to calculate relative ratios of the indicated protein over loading control (β-Actin), which were then normalized based on the corresponding ratio in the wild type control sample.

## Real-time PCR

Total RNA was isolated from the tissues using Trizol (Invitrogen). cDNA was then reverse-transcribed from the total RNA using MultiScribe RT kit (Thermo Fisher). Real-time PCR was performed on Applied Biosystems StepOne Plus machine. *Cdkn1a (p21)*, *PTEN* and *actin* genes were detected using predesigned commercial TaqMan primers for each gene accordingly (*Cdkn1a*: Mm00432448-m1, *PTEN*: mm00477210-m1, and *Actin*: Mm00607939-s1). Results were calculated using the $\Delta\Delta C_T$ method and then normalized to actin.

## Statistics

Data were expressed as the mean ± SEM. Statistical analyses were carried out with GraphPad Prism version 8.0 or Microsoft Excel. Significance in the Kaplan-Meier analyses to determine and compare the progression-free survival was calculated using the log-rank test. The mass spectra were analyzed by MAVEN software and the peak area of each detected metabolite was obtained. Statistical significance of metabolites was determined by a paired two-tailed Student's t-test, and five mice from each genotype were used. p-Value of <0.05 was considered statistically significant. The heatmap of

amino acids was generated by using R 3.6.1. program, and all values are processed by the mean normalization. Pearson algorithm was used for the hierarchical clustering of the rows.

## Acknowledgements

We are grateful to Dr. Eileen White for her advice during the preparation of the manuscript; Wenping Wang in the Guo laboratory for helping with generation of heat map; Amy Lee, Nuha Syed, Akash Raju, Jerry Kong and Enrique Lopez in the Guo laboratory for helping with mouse ear tagging and genotyping. This work was supported by National Institute of Health (NIH) grant R01 CA237347-01A1, NIH grant K22 CA190521, American Cancer Society grant 134036-RSG-19-165-01-TBG, GO2 Foundation for Lung Cancer, the Lung Cancer Research Foundation and Rutgers Busch Biomedical grant to J.Y.G; New Jersey Commission on Cancer Research (NJCCR) grant DFHS18PPC021 to K.K; NJCCR grant DCHS19PPC013, a scholarship from the Cox Foundation for Cancer Research and Mistletoe Research Fellowship to V.B; and NIH P30 CA072720 to Rutgers Cancer Institute of New Jersey.

## Additional information

### Funding

| Funder | Grant reference number | Author |
| --- | --- | --- |
| National Cancer Institute | R01CA237347-01A1 | Jessie Yanxiang Guo |
| National Cancer Institute | K22 CA190521 | Jessie Yanxiang Guo |
| American Cancer Society | 134036-RSG-19-165-01-TBG | Jessie Yanxiang Guo |
| GO2 Foundation for Lung Cancer | Young Innovators Team Awards | Jessie Yanxiang Guo |
| Lung Cancer Research Foundation | Research Grant | Jessie Yanxiang Guo |
| New Jersey Commission on Cancer Research | DFHS18PPC021, Postdoc fellowship | Khoosheh Khayati |
| New Jersey Commission on Cancer Research | DCHS19PPC013, Predoctoral fellowship | Vrushank Bhatt |
| Rutgers Busch Biomedical Grant | Research Grant | Jessie Yanxiang Guo |
| Cox Foundation for Cancer Research | Predoctoral fellowship | Vrushank Bhatt |
| Mistletoe Research Fellowship | Predoctoral fellowship | Vrushank Bhatt |

The funders had no role in study design, data collection and interpretation, or the decision to submit the work for publication.

### Author contributions

Khoosheh Khayati, Data curation, Formal analysis, Validation, Visualization, Methodology, Writing - original draft, Writing - review and editing; Vrushank Bhatt, Zhixian Sherrie Hu, Sajid Fahumy, Xuefei Luo, Methodology; Jessie Yanxiang Guo, Conceptualization, Resources, Data curation, Formal analysis, Supervision, Funding acquisition, Validation, Investigation, Visualization, Methodology, Writing - original draft, Project administration, Writing - review and editing

### Author ORCIDs

Khoosheh Khayati (iD) https://orcid.org/0000-0003-2424-837X
Jessie Yanxiang Guo (iD) https://orcid.org/0000-0001-9212-7954

### Ethics

Animal experimentation: This study was performed in strict accordance with the recommendations in the Guide for the Care and Use of Laboratory Animals of the National Institutes of Health. All of

the animals were handled according to approved institutional animal care and use committee (IACUC) protocols (#I15-074) of the Rutgers University.

## Decision letter and Author response

Decision letter https://doi.org/10.7554/eLife.62377.sa1
Author response https://doi.org/10.7554/eLife.62377.sa2

## Additional files

### Supplementary files

• Source data 1. Levels of metabolites in the sera of mice measured by LC_MS.

• Source data 2. Levels of metabolites in the intestine of mice measured by LC_MS.

• Transparent reporting form

### Data availability

All data generated or analysed during this study are included in the manuscript and supporting files.

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
