## [Decision Letter]

**Acceptance summary:**

This paper shows that deletion of STK11 (also known as LKB1) in adult mice induces compensatory activation of autophagy to promote tissue homeostasis and animal survival. These results are contradictory to our expectation that STK11/LKB1 should inhibit autophagy as STK11/LKB1 is known to activate AMP-activated protein kinase (AMPK), a positive regulator of autophagy. Thus, this study provides important insights into the role of STK11/LKB1 in physiological settings in vivo, especially in terms of autophagy regulation.

**Decision letter after peer review:**

Thank you for submitting your article "Autophagy Compensates for Lkb1 Loss to Maintain Adult Mice Homeostasis and Survival" for consideration by *eLife*. Your article has been reviewed by three peer reviewers, including Noboru Mizushima as the Reviewing Editor and Reviewer #1, and the evaluation has been overseen by David Ron as the Senior Editor. The following individual involved in review of your submission has agreed to reveal their identity: Boyi Gan (Reviewer #2).

The reviewers have discussed the reviews with one another and the Reviewing Editor has drafted this decision to help you prepare a revised submission.

Summary:

This study shows that, if LKB1 is conditionally deleted in adult mice, autophagy is upregulated. This could be a compensatory mechanism because the lifespan of Lkb1 KO mice is shortened by additional KO of ATG7 (from 6 to 4 weeks). The authors further demonstrate that the intestinal barrier function is impaired after acute deletion of LKB1. This phenotype can also be partially rescued by autophagy through inhibition of p53 induction. Overall, this is an interesting and solid mouse genetic study, which significantly expands our understanding of LKB1 and autophagy in regulating tissue homeostasis and animal survival in vivo. One particular novel aspect of the study is that, while it is believed that LKB1 KO should inhibit autophagy by inactivating AMPK, this study showed convincingly that, in response to Lkb1 deficiency in vivo, autophagy is actually activated to promote tissue homeostasis and animal survival. This study, therefore, highlights the critical need to conduct rigorous in vivo genetic studies to study the LKB1 function.

1) How Lkb1 deletion leads to autophagy activation in vivo remains unclear. The authors mentioned that, besides LKB1, other AMPK kinases, such as CaMKK2 and TAK can phosphorylate AMPK to activate autophagy. However, it seems unlikely that Lkb1 deletion would even increase AMPK phosphorylation in vivo (based on their observation that autophagy is increased upon Lkb1 deletion in vivo). In Figure 1B, authors need to check phosphor-AMPK and AMPK in various tissues of Lkb1 WT and KO mice.

2) Related to the above comment, one of the key messages in this study is that the Lkb1-AMPK-mTORC1 axis may not be critical in autophagy regulation. However, the activity of both AMPK and mTORC1 is not monitored in this study. The authors show that acute depletion of Lkb1 reduced the levels of amino acids and other major metabolites. Amino acid deficiency is expected to inactivate mTORC1 in an AMPK-independent manner, leading to autophagy induction. Therefore, it is important to determine whether mTORC1 is inhibited in Lkb1, ATG7, and double KO mice. If the activity of neither AMPK nor mTORC1 is changed, it is possible that the increased autophagy observed in various tissues of Lkb1 KO mice more likely reflects a systemic adaptive response to the metabolic collapse. Obviously, it will be challenging to formally test this hypothesis, but the authors should at least discuss this or other potential mechanisms which underlie increased autophagy phenotype in Lkb1 KO mice.

3) The authors conclude that autophagy is activated in Lkb1 KO mice solely based on the HCQ treatment experiments (Figure 1D). As this is one of the critical points in this study, autophagy induction should be confirmed by other methods, such as by checking phosphorylation of ULK1-S757 (PMID: 21258367) and ATG16L1-S278 (PMID: 31768061).

4) Despite the marked disruption of the intestinal barrier in the Lkb1-deficient mice (Figure 3F), a thorough study on epithelial cell death was missing. In addition to caspase-3 (Figure 3E), Tunel assay and other types of cell death should be evaluated. Also, the authors need to check whether p53 KO rescued the increased cleaved caspase-3 staining in triple KO mice.

Revisions expected in follow-up work:

While the elevation in autophagy extended the lifespan of Lkb1-KO mice, it failed to restore their homeostasis and survival. Also, although p53-deficiency improved the symptoms imposed by *Atg7*-deficiency, it did not ameliorate any defects caused by Lkb1-deficiency alone. Thus, it is important to understand the autophagy-independent pathway impaired by LKB1 loss and how it contributes to the dysfunctions caused by LKB1 deficiency. Although the authors could address these issues in the future, at least discuss these autophagy-independent roles of Lkb1 in more depth in this study.

[Editors' note: further revisions were suggested prior to acceptance, as described below.]

Thank you for resubmitting your work entitled "Autophagy Compensates for Lkb1 Loss to Maintain Adult Mice Homeostasis and Survival" for further consideration by *eLife*. Your revised article has been evaluated by David Ron as the Senior Editor and a Reviewing Editor.

The manuscript has been improved but there are some remaining issues that need to be addressed before acceptance, as outlined below:

The authors monitor the level of ATG12-ATG5 as an indicator of autophagic activity. However, ATG12-ATG5 is not degraded by autophagy and is not generally considered as an autophagy indicator. To avoid confusing readers, the authors may consider removing the data in Figure 1G and the corresponding text.

---

## [Author Response]

Revisions for this paper:1) How Lkb1 deletion leads to autophagy activation in vivo remains unclear. The authors mentioned that, besides LKB1, other AMPK kinases, such as CaMKK2 and TAK can phosphorylate AMPK to activate autophagy. However, it seems unlikely that Lkb1 deletion would even increase AMPK phosphorylation in vivo (based on their observation that autophagy is increased upon Lkb1 deletion in vivo). In Figure 1B, authors need to check phosphor-AMPK and AMPK in various tissues of Lkb1 WT and KO mice.

As suggested, we checked the p-AMPK(T172) and the phosphorylation of its downstream targets ULK1 and ACC in different tissues by IHC. pAMPK (T172), pULK1(S555) and pACC (S79) were observed in the tissues of *Stk11^-/-^* mice, indicating that AMPK can be activated by other kinases if *Stk11* was deleted. Interestingly, although pAMPK was not further increased in the tissues of *Stk11^-/-^* mice, we observed that pS6 which indicates the activation of mTORC1 signaling was significantly reduced in the tissues of *Stk11^-/-^* mice compared with WT mice. Moreover, the level of pULK1(S757), a phosphorylation site by mTORC1, was reduced in the tissues of *Stk11^-/-^* mice compared with WT mice. This results also suggest an AMPK-independent mTORC1 activation. These new data were added in the revised manuscript (new Figure 1E and F, and new Figure 1—figure supplement 1, Results and discussed more in the Discussion.

2) Related to the above comment, one of the key messages in this study is that the Lkb1-AMPK-mTORC1 axis may not be critical in autophagy regulation. However, the activity of both AMPK and mTORC1 is not monitored in this study. The authors show that acute depletion of Lkb1 reduced the levels of amino acids and other major metabolites. Amino acid deficiency is expected to inactivate mTORC1 in an AMPK-independent manner, leading to autophagy induction. Therefore, it is important to determine whether mTORC1 is inhibited in Lkb1, ATG7, and double KO mice. If the activity of neither AMPK nor mTORC1 is changed, it is possible that the increased autophagy observed in various tissues of Lkb1 KO mice more likely reflects a systemic adaptive response to the metabolic collapse. Obviously, it will be challenging to formally test this hypothesis, but the authors should at least discuss this or other potential mechanisms which underlie increased autophagy phenotype in Lkb1 KO mice.

We thank the reviewers’ valuable comments and suggestion.

As mentioned above, we found that the level of pS6, a readout of mTORC1 signaling was reduced in the tissues of *Stk11^-/-^* and *Atg7^-/-^*;*Stk1^-/-^* mice although pAMPK was not further increased when *Stk11* was systemically deleted. mTORC1 activity can be stimulated by growth factors and cellular energy through inhibition of the TSC complex (TSC1-TSC2-TBC1D7), a negative regulator of mTORC1. However, amino acids can also signal to mTORC1 independently of the TSC complex via sensors localized on lysosomal membrane. We found that systemic ablation of *Stk11* and autophagy led to the depletion of amino acids and other essential metabolites in serum, including leucine, arginine and methionine which are essential for mTORC1 activation. Therefore, systemic metabolic collapse could also lead to mTORC1 inactivation. Our study suggests that the inactivation of TORC1 in *Stk11^-/-^* mice could come from both AMPK dependent and independent pathway, thereby activate autophagy to temporally compensate for acute Lkb1 loss for mouse survival.

It’s challenging for us to formally test above hypothesis, but we have discussed it in revised Discussion.

3) The authors conclude that autophagy is activated in Lkb1 KO mice solely based on the HCQ treatment experiments (Figure 1D). As this is one of the critical points in this study, autophagy induction should be confirmed by other methods, such as by checking phosphorylation of ULK1-S757 (PMID: 21258367) and ATG16L1-S278 (PMID: 31768061).

As mentioned above, we confirmed the autophagy activation signaling by measuring pAMPK (T172), pULK1 (S757), pULK1(S555), pS6 (S235/236) and Atg5-12 conjugation. When mice were treated with HCQ to block autophagy flux, besides p62 and LC3II accumulation, we also observed that Atg5-12 conjugated form was increased in the tissues of *Stk11^-/-^* mice compared with WT mice, indicating autophagy activation. These new data were added to new Figure 1E, F and G, Results.

4) Despite the marked disruption of the intestinal barrier in the Lkb1-deficient mice (Figure 3F), a thorough study on epithelial cell death was missing. In addition to caspase-3 (Figure 3E), Tunel assay and other types of cell death should be evaluated. Also, the authors need to check whether p53 KO rescued the increased cleaved caspase-3 staining in triple KO mice.

As suggested, epithelial cell death was evaluated. In addition to the apoptotic cell death observed in the tips of villus as we showed in previous submission, we also observed more apoptotic cell death in intestinal epithelial cells in *Atg7^-/-^*;*Stk11^-/-^* mice compared with WT, *Atg7^-/-^* and *Stk11^-/-^* mice examined by cleaved caspase3, which was added in the new Figure 3E, Results. We also performed TUNEL assay in the intestine tissues. As expected, TUNEL staining was significantly increased in the intestine of *Atg7^-/-^*;*Stk11^-/-^* mice compared with WT, *Atg7^-/-^* and *Stk11^-/-^* mice (new Figure 3F and Figure 3—figure supplement 1E, Results), which is consistent with cleaved caspase 3 IHC staining. We also evaluated the apoptosis in mice with *Trp53* deletion by cleaved caspase 3 IHC and TUNEL staining. Apoptotic cell death was abolished when *Trp53* was systemically deleted in triple KO: *Atg7^-/-^*;*Stk11^-/-^*;*Trp53^-/-^* mice (new Figure

4D and E, Results and Discussion).

Additionally, we assessed the possibility of other types of cell death in the intestine of *Atg7^-/-^*;*Stk11^-/-^* mice. Ferroptosis is initiated by the failure of the glutathione-dependent antioxidant defenses, resulting in unchecked lipid peroxidation accumulation and eventual cell death. We performed IHC for 4-hydroxynonenal (4-HNE), a marker for lipid peroxidation. We found that staining of 4HNE was significantly increased in the intestine of *Atg7^-/-^*;*Stk11 ^-/-^* mice compared with WT, *Atg7^-/-^* or *Stk11^-/-^* mice, indicating that ferroptosis could occur when Atg7 and *Stk11* are concurrently deleted in adult mice. However, more investigation needs to be performed to validate that ferroptosis is induced in *Atg7^-/-^*;*Stk11^-/-^* mice, which is beyond the scope of current study.

Revisions expected in follow-up work:While the elevation in autophagy extended the lifespan of Lkb1-KO mice, it failed to restore their homeostasis and survival. Also, although p53-deficiency improved the symptoms imposed by Atg7-deficiency, it did not ameliorate any defects caused by Lkb1-deficiency alone. Thus, it is important to understand the autophagy-independent pathway impaired by LKB1 loss and how it contributes to the dysfunctions caused by LKB1 deficiency. Although the authors could address these issues in the future, at least discuss these autophagy-independent roles of Lkb1 in more depth in this study.

We thank the reviewers’ valuable comments and suggestion.

We discussed autophagy-independent pathways that are impaired by Lkb1 deficiency in adult mice for follow-up in the revised manuscript (Discussion).

[Editors' note: further revisions were suggested prior to acceptance, as described below.]

The manuscript has been improved but there are some remaining issues that need to be addressed before acceptance, as outlined below:The authors monitor the level of ATG12-ATG5 as an indicator of autophagic activity. However, ATG12-ATG5 is not degraded by autophagy and is not generally considered as an autophagy indicator. To avoid confusing readers, the authors may consider removing the data in Figure 1G and the corresponding text.

Thanks for your comments and suggestion. We removed the data in Figure 1g and the corresponding text.